# HBIM Meta-Modelling: 50 (and More) Shades of Grey

**Martina Attenni, Carlo Bianchini, Marika Griffo *** and Luca James Senatore

Department of History, Representation and Restoration of Architecture, Sapienza, University of Rome,
00186 Rome, Italy
* Correspondence: marika.griffo@uniroma1.it

**Abstract:** The paper aims at investigating modelling strategies in HBIM context to identify at what extent the final use of the model might affects, or should affect, the modelling approach itself. Moreover, the discussion wants to shed light on the possibility of connecting in just one digital environment several instances connected to the building. These aims will be discussed presenting and evaluating two different modelling approaches: the "black box" modelling and the "white box" model-ling. The two terms are partially borrowed from computer science to explain two types of testing. The "black box" testing is performed without any preliminary knowledge about the system functionality and internal components; on the contrary, the "white box" testing, implies a full knowledge of the system. These two approaches will be compared to two ways of conceiving a building information model. In conclusion, the paper will investigate the possibility to integrate in just one model, the grey box model, the two ones previously discussed.

**Keywords:** grey box testing; HBIM as built; ideal model; HBIM meta-modelling

## 1. Introduction

The built heritage conservation process might take advantage from the construction of an ontologically consistent digital 3D model. This is true if this model is able to describe, in the digital world, an architectural manufact and its characteristics. In other words, if this model is a digital twin of the building. The idea of the "twin" was developed in the military field [1], and then, it was taken up again referring to HBIM [2]. Around this concept, several researches and experimentations have been carried out to investigate what exactly the digital twin is and how to build it. Right now, as often happens with the development of new disruptive approaches, although the principles are consolidated and shared, the operational methods are not yet fully defined. The question is which approach should be used to create the Twin.

From the critical interpretation of the raw data (point cloud), which describe the morphometric characteristics of a building, it is possible to produce an indefinite number of models. Each of these models is informed by considering only a limited set of data. This imposes certain a priori choices with respect to the modelling process, and whether it is to be similar to reality or a representation of the ideal form. Each model is a kind of interpretation of the starting data, the modeler must decide whether to follow an "ideal model" approach or with an antithetical approach, to build the model as a uncritical digital reconstruction of the current state. The proposed research has its novelty in finding a solution to overcome the existing duality in modelling approach to reach a complex model which is structured to contain several partial models.

In the field of computer science the consistency of an algorithm is verified a posteriori through two modes of testing [3,4] defined black and white: the "black box" examine the functionality of an application without entering into its internal structures or its operation; the "white box" examine the internal structures or the operation of an application having available all the system data. Even in the case of architecture, the modeling process is implicitly a recursive testing operation, performed on the available database. This operation

of testing/modeling can be approached according to two levels of reading, one directly conditioned by the data, and one guided by a critical approach that seeks ideality: the "black box" modelling and the "white box" modelling. Both are based on a structure to which models must adhere, called meta-modeling [5,6]. In the field of HBIM, metamodeling describes the syntax of the models through the definition of aggregation rules of the elements that can help to define the semantics of whole model according to its objective and its use [7].

Starting from these considerations, the contribution will show how it is possible to construct the two different models of an existing buildings, highlighting their weakness and strengths. Both solutions correspond to the digital representation of a specific amount of data, providing a specific point of observation. In conclusion, the paper will investigate the possibility to integrate in just one model, the grey box model, the two ones previously discussed.

## 2. Background

Existing buildings do represent a hard subject to tackle under several points of view. We could summarize the problem observing that they keep undisclosed most of the information about their inner nature, structure, and consistency so that all actors dealing with their study or transformation (restoration, retrofitting, rehabilitation, etc.) must address this issue and continuously try to fill the gap between the searched and found information [8].

Researchers, scholars, and designers are quite familiar with this state of play that is assumed as a constraint by now, a sort of red line to coexist with during any project involving existing buildings [9].

Digital technologies have deeply changed this consolidated scenario thanks to a wide "bundle" of hardware and software tools paving the way to a renovated interaction between our real world and its abstract versions [10]. On one side, terrific progress have been made in acquiring data from objects [11]; on the other, the digital modelling software has provided new paradigms for reconstructing them, for interacting with them, for simulating their possible transformations [12].

Core of this multifaceted set of activities is the Model, namely the virtual simulacrum of a real element, which represents the goal and the medium of this interaction [13].

While for many years now the Model has been a dominant topic of investigation (and concern) for scholars and technicians, lately the attention has been shifting towards the Modelling. In other words, the focus has been moving from the final output to the set of decisions and actions guiding the process of construction and information (in the sense of defining an informative content) of all digital elements composing, together, the final output [14].

Under this perspective, modelling buildings certainly implies constructing, that is to say, applying a set of logical and practical rules not afar from those that typically govern a real construction site [15]. In fact, as in the construction workflow we must break down the building into an organised catalogue of elements and arrange the site and the sequence of works, likewise the construction of a 3D Model must approximately respect similar steps and constraints. In a BIM process, the identification and generation of BIM digital objects is generally called Semantic Segmentation [16].

In this framework, the construction of 3D Models is a real heuristic activity: differently from graphic 2D models (drawings), in a 3D Model any portion of the object must be defined as anything can become visible and anything is visible during its exploration. Furthermore, buildings can be regarded as a coordinated set of basic elements respecting design and construction patterns quite predictable and manageable [17]. Finally, any 3D element must be augmented with additional content in order to show not only its quantitative properties (i.e., geometry) but also its qualitative ones (material, physical parameters, performance, etc.).

BIM systems provide the most promising digital environment for hosting, organizing, and interacting with all this information with the additional benefit that the graphic interface provided by the software could highly improve the access to the building's

informative database. For these (and other) reasons the Building Information Modelling has increasingly become a standard reference for the many actors involved in the Architecture, Engineering and Construction (AEC) Industry. This complex process (that in addition is implying the transition from CAD to BIM systems) had in the beginning a very effective impact especially on new constructions with the wider ambition of extending its paradigms to the existing ones.

For some years, the application of BIM to existing buildings and particularly to those displaying a strong historic or cultural value (i.e., Built Cultural Heritage-BCH) has been considered either a mere switch of procedures from new buildings to existing ones, or a digital environment technical problem. As "buildings are buildings" and BIM has proven highly successful in terms of time and resources optimization for new ones, these advantages were considered as extendible to the existing stock without much effort.

We can now affirm that this approach has revealed being too simplistic and actually unable to handle the complexity of BCH.

As a result, investigations in this field have been encouraged at national and international level [18] for its relevant impact on AEC Industry being the interventions on existing buildings already the most relevant fraction of the whole amount [19].

The picture emerging from these activities has outlined many inconsistencies while attempting to apply the new buildings' BIM standards to the existing ones. Besides, far from being merely technical, these issues have proven to address directly the conceptual and cultural background underneath the modelling phase. The "H" standing for "Heritage" currently accepted before the acronym BIM is an outcome of last years' efforts and intends to mark the class of objects involved in the process as well as the specificity of the workflow to be adopted.

Moreover, our discussion must start from the acronym BIM. "Building" of course refers to the objects addressed and their constructive workflow. "Information" refers instead to the ability of the system to manage information connected to these objects. "Model", finally, corresponds to the digital double of the Building considered (sometimes named digital twin), which represents the scaffolding to which "Information" is stitched as well as the goal and medium of any interaction between the real and the virtual version of the object itself.

According to the conceptual framework we discussed in the previous paragraphs, it is quite clear why BIM works well with new buildings: at least in principle, in fact, information about both the elements and their assembling can be known in advance. The high correspondence between the virtual model and the real object also allows for highlighting inconsistencies and interferences at a very early stage of the construction workflow. This acknowledged property produces the most significant advantages in terms of efficiency and cost reduction.

Existing buildings, unlike new ones, keep instead undisclosed most of the information about their inner nature, structure and consistency making in this case the parametric and informative modelling of BCH much harder, both in terms of geometric transposition of the real world and of its qualitative and semantic description. These difficulties are also coupled to the essential inelasticity of BIM parametric modellers, subjected to digital "libraries" of objects that predictably would clash against the holistic character of BCH, especially when it is layered or deteriorated.

In this framework, some questions appear to be crucial: being the BIM model made of solid elements how can we go beyond the surface of objects and capture their inner 3D structure? How can we set semantic and aggregation rules for BCH digital objects? How can we define the limit between evidence and subjective hypothesis in the modellers' work? In view of extensive application of BIM workflows to AEC processes, how BIM models can be entrusted as legal documents in the tendering and contractual procedures?

Before discussing these matters, we must preface that we propose to push beyond the limits of the technical approach recalled at the beginning of this paper, coupling it with a humanistic one. The reason for this choice bases on the evidence that the majority of

HBIM Models display a very high technical performance but conversely a poor consistency in terms of "reading" of the built object. We have so assumed in all projects presented in the following paragraphs, that the ingredient missing in the first case is semantic, i.e., the ability to correctly recognize, interpret and synthetize the different elements of any fabric as well as its composing rules. This last remark should not sound like a novelty, as Architecture intrinsically embeds the concept of "language" at least since the Vitruvian firmitas, utilitas, venustas, where structural, functional, and decorative elements contribute to make its words, sentences and chapters.

Not neglecting (nor underestimating) the technical aspects, we thus simply counterbalanced them with the strong conceptual structure coming from multi-layered architectural investigations (compositional, historical, constructive, structural, etc.) aiming at providing tools for a deeper and more consistent modelling of BCH.

### 2.1. Point Clouds and Informative Content

The first task we dealt with proceeding along the HBIM workflow was the digitalization process. For objects, including existing buildings, digitalization seems not to represent an issue anymore. 3D capturing technologies either structured (3D scanning) or unstructured (Structure from Motion) are in fact by now commonplace. Besides, BIM software can now import huge set of numeric data into any BIM environment and directly use it for modelling digital objects [20].

Despite this "brute force" displayed by hw/sw sstems, still there is a lack of tools and methods for assessing the quality of point clouds. Sometimes the absolute number of points composing the cloud is assumed as a parameter; in other occasion it is the cloud density (number of point/reference area) but the truth is that on one hand there is no accepted rule and on the other those that sound as rules deceive instead a deep inconsistency.

Nevertheless, the availability of high-quality numeric models is a key factor for producing consistent HBIM Models of BCH. The measuring of this quality certainly de-pends on the number of acquired points but this parameter risks to be meaningless un-less it is coupled with the information potentially retrievable from that dataset [21].

The strategy adopted in our projects starts from the traditional concept of "scale" we use for drawings where the notation 1:50 or 1:100 is still a common reference for both the maker and the reader of the drawing. This dimensional reference would tell the former "how much" he/she would/could detail the drawing, while show to the latter "how much" he/she would/could extract from the drawing in terms of measure. This evidence, far from being merely conventional, is on the contrary thoroughly scientific, even if based on statistics: human beings, in fact, cannot distinguish lines closer than 0.2/0.3 mm and this length at drawing level corresponds to 1/1.5 cm at 1:50 scale and to 2/3 cm at 1:100. It makes no sense, thus, for the maker to detail his/her representation beyond this threshold and for the reader to expect more detail than that corresponding to the scale.

The application of this concept to point clouds is easy and consistent at the same time. It is paramount how the "density" of the point cloud represents the key parameter for identifying the "scale" of a point cloud. In fact, considering a point cloud with a density of 1 point every 5 mm, we could assume this interval as the reference distance in the "real world" to which the scaled distance of 0.2/0.3 mm corresponds to a scale between 1:10 and 1:20. The adoption of this method has been quite beneficial for our projects. On one side it has allowed for a better design /implementation of the 3D capturing campaign (especially the SfM's one) because the established reference scale has led to more uniform point clouds; on the other, especially in the 2D modelling phase, it has "obliged" the modellers to stick to the required informative content consistent with the data acquired.

### 2.2. Semantic and Aggregation Rules for BCH Digital Objects

We already mentioned that for new constructions, we can count on objects and assembling rules that are established during the BIM modelling/informative phases in accordance with logical/operational pathways specific of the building sector [22]. The overall goal

is to highlight geometric inconsistencies (clash detection) and the non-compliance of the informative model with specific reference standards (code checking).

The application of these rules to BCH is still erratic. Differently from new constructions where the information attached to each digital object is consistent and known in advance, an existing object permanently hides most of its information.

For this reason, the correspondence between any existing building and its digital counterpart is highly jeopardized compromising the HBIM workflow. One of the core problems is connected to the concept of "Ideal Model". While in new constructions the Ideal Model coincides with the coordinated set of the technological elements established "from scratch" by the designer, for existing ones this approach is evidently not applicable even if any existing object in a certain stage of its history has been a "new one".

Could we have access to the ideal model of that building we could then deploy the full potential of BIM approach. The methods through which we could tend to the reconstruction of such a model imply deducting the original building's design, not only in terms of geometry but also from a compositional, technological, material, constructive and evolutionary standpoint. Here is the task where the humanistic approach we mentioned in the previous paragraphs is most beneficial: in our experience, in fact, the 3D data coming from the building are to be complemented with 1D (texts) and 2D (drawings, pictures) information normally stored in archives, libraries, collections, etc. This is the only tool we have to "go beyond" the skin of an existing building [23].

The Ideal Model we have been describing is in most cases only the initial one. According to an "additive" approach, in fact, such modelling phase should be able to incorporate new information into the BIM model as additional layers that overlay, intersect, or substitute original structures. This process, that for BCH normally embraces a period of centuries, ends with the so called "BIM-as-is" describing the present condition and consistency of the artefact.

Other issues are instead related to the international (and national) standard parameters used in the BIM and HBIM process: the Level of Development (LOD) and the Level of Detail (LoD). Whereas the first intends to "measure" the reliability of the information characterizing a BIM model, the second defines the graphic detail of digital objects in case of visualization or representation. BIM objects are also defined not only by geometric characters but also by the informative ones, referred as Level of Information (LOI). In Italy, the norm ISO 19650-1 of 2018 replaced the Levels of Information (LOI) with the Level of Information Need (LOIN), which is further calculated by combining geometric requirements (LOG, Level of Geometry), and non-geometric ones(LOI, Level of Information) [24].

Current BIM authoring software provides tools to model digital objects with different graphic detail; therefore, there it seems to be a close analogy between LOD and LoD so that we can easily conclude that simply increasing the detail of an object we can pass from one LOD to another assuming the Level of Development as equivalent to its graphic details. On the contrary, LODs are independent from LoDs and although they have been adopted to provide a standard for digital objects' consistency, still they do not take into account the quality of the information on which they are constructed [25].

### 2.3. Documental Value of HBIM Models

For more than 10 years now, the AEC Industry has been increasingly switching to BIM to optimize processes and reduce costs. Sharing the same interest, decision-makers have quickly adopted a similar policy for public tenders, both at UE (Directive 2014/24/EU) and member states level. Accordingly, since the DM MIT 560/17 the Italian legislation has progressively introduced the usage of BIM as a constraint in all public procurements. Currently, all works above 5 Mln€ must adopt BIM approach and that threshold will reduce to 1 Mln€ in 2023.

Despite this political will, both Italian AEC Industry and Public Administrations seems to belong to the laggard group described by the Roger's technology adoption curve [26,27]. Many concurrent reasons produce obviously this phenomenon (i.e., the lack of familiarity

of professionals and officers, the rigidity of BIM systems but also the semantic issues described so far) but one seems to have a potential strong impact on all segments of the BIM and HBIM workflow.

The key element in this case is "interoperability", i.e., the crucial property that BIM and HBIM Models must display to allow all actors involved in the process to work "in parallel" instead of following the current "in series" pipeline. Even if this "process innovation" is far from having become commonplace, nevertheless it has been agreed to be the actual booster component for a full deployment of BIM potentials.

The idea is simple: instead of having the different actors and stakeholders in the building process providing their contributions according to a "one to one" scheme, interoperability implies that all could work concurrently and in real time on the same shared model. Apparently, this feature seems to be automatic, a sort of emerging property of BIM systems due to the integration between the informative and the geometric components of the model but on the contrary it is not. While on one hand BIM systems can rely on a sound digital infrastructure that seems flexible and wide enough to respond, in perspective, to the needs brought in by interoperability, on the other, there is still great uncertainty on how to manage the access policies of the different BIM Model users.

The place where all these activities would take place is the so-called Common Digital Environment (CDE), i.e., that shared "sandbox" where users interact with both the model and the collected information. Despite the current rudimental exchange rate between the different BIM software, still it is paramount that interoperability implies registering, keeping trace and monitoring the different activities developed on the model. It is not a matter of generating a typical "log file" or updating the metadata of the model. We should instead consider the problem from a pure digital standpoint in terms of integrity of the model while subjected to its progressive evolution. In the end, in fact, an HBIM Model is a file or a coordinate collection of files and the "integrity of the Model" corresponds to the integrity of these files. In other words, if we want to design a method to preserve the different qualities of an HBIM Model, not only we must improve the exchange of information from one format to another, but also act on its digital codification, i.e., the files.

A general method to ensure this crucial feature is protection, a term we all consider familiar (for good or bad) when interacting with the digital world. Many technologies enforce protection, generally exploiting encryption protocols commonly highly successful for the majority of one-to-one communications.

This way of protecting the Model's coding is not enough in an interoperable environment like CDE: we must switch in fact from a one-to-one approach to a multi-peer one where the "trust" in the Model does not depend on the reputation of the sender but must be embedded into the models themselves. Among the various technologies developed to solve this general problem, Blockchain [28] seems to be particularly promising.

In view of a full deployment of HBIM systems, the same Blockchain algorithms are expected to provide more consistent solutions for another key issue: validation and certification of Models (possibly also from a legal standpoint).

Despite BIM and HBIM models are dramatically increasing in number (not necessarily in quality as we said) under the pressure of legislation, nevertheless they still have little or zero legal value. In fact, the majority of tenders still base on digital versions of traditional physical documents (like pdf files) and mention BIM models only as "as-built" to be delivered at the end of the work just to comply with legislation. This approach is not only resources and time consuming but also opposite with respect to the aims of the policy makers. Therefore, it is urgent to investigate the methods for supporting a full BIM transition in the AEC Industry.

## 3. Methods

The following two subsections aim at make explicit the procedure that leads from the 3D documentation to the final modelling. This procedure considers two different approaches to the HBIM: the "black box" modelling (Section 3.1) and the "white box"

modelling (Section 3.2). The "black box" modeling is performed considering data derived by 3D integrated survey, without any preliminary analysis about the architectural design project; on the contrary, the "white box" modelling, implies a full preliminary analysis of the design. In both cases, anyhow, the accuracy of each model element and the correspondent LOD must always be explicit [29].

### *3.1. "Black Box" Modelling of the Building in Piazza Borghese, Rome*
3.1.1. The Building Historical Stratification

The building that currently hosts the faculty of Architecture, Sapienza, University of Rome, comes from the annexation into one of several structures. All these structures belonged to the Borghese family and were used for secondary functions connected to the main property, the Palazzo Borghese, located in the same square, piazza Borghese. This strong transformation was conducted between 1923 and 1928, some portions of the old buildings were demolished, some others were maintained and new connection elements, such as the external facades and the main staircase, were added. In 1928 the new building was inaugurated as the site for the Regio Istituto Superiore di Scienze Economiche e Commerciali then turned, in late '30 into the faculty of Economics building. This function affected the technological and architectural solutions adopted for the spaces to gradually adapt them to the required performances and standards. As a result, this building owns a strong stratification, from the chronological point of view, and a complex fruition system, with several external accesses and articulated spaces configuration.

Later in '70, the building was assigned to the faculty of Architecture of Sapienza and, following this, new interventions were carried out from 1970 to 1977 to adapt the existing spaces to the new functions, inserting new lecture and study rooms (Figure 1).

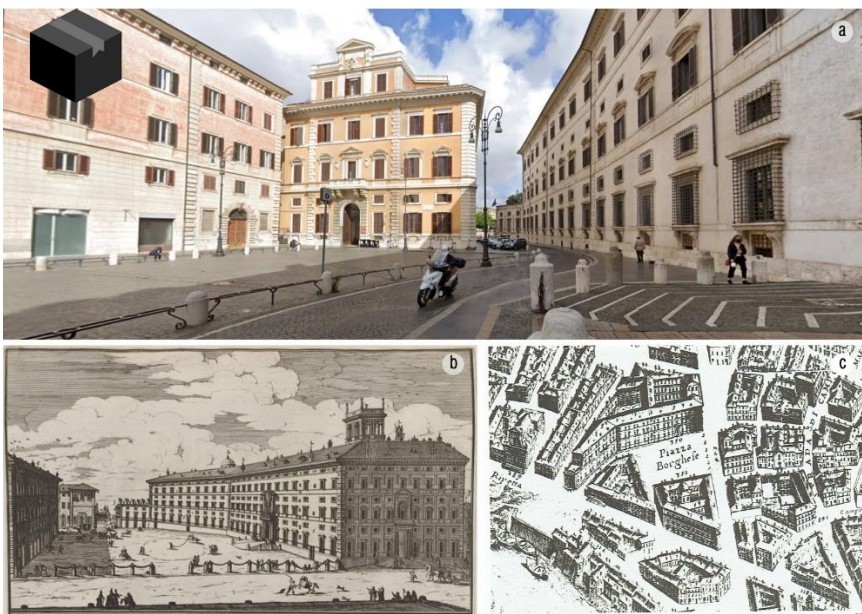

**Figure 1.** (**a**) The building in Piazza Borghese in its current state. (**b**) Piazza Borghese as represented in a G. B. Falda drawing, 1943. (**c**) Piazza Borghese in a planimetric view of Rome, made by Giuseppe Vasi in 1676.

3.1.2. Architectural Survey for HBIM Purposes

As for the other properties of Sapienza University real estate, since 2021, the building in Piazza Borghese has undergone a digitalization process aiming at collecting an extensive 3D survey and BIM of all the buildings. In this framework, which is fully related to Heritage–BIM more than the classic BIM, the goal is to have a 3D model able to support the decision-making process in terms of maintenance and refurbishing. Given this, the HBIM must consider issues related to the architectural aspects, the Mechanical, Electrical

and Plumbing (MEP) aspects and the structural ones. This means gathering specialistic information related to the building in its current form, based on the data provided by a 3D survey point cloud integrated with annotations coming from the different fields of investigation involved. Considering the primary scope of this digitalization process, the adherence of the HBIM model to the state of the art of the building was evaluated as a prerogative characteristic. In this sense, survey activities have been designed and carried out to ensure, in each phase, a high level of metric, geometric and informative accuracy.

The entire survey process has been set as a sequence of four phases: bibliographic research, data capturing, data elaboration, data check and validation. The first step was to collect bibliographic information to provide a preliminary list of architectural elements evaluated as of high quality and identifying the building itself (Figure 2). This list was then completed and optimized thanks to on-site inspections and, for each of these elements, a small description, a picture, and a code has been provided. The second phase, the data capturing, has been performed integrating a laser scanner survey (RTC 360 by Leica Geosystems, Rome, Italy) of all the interior and exterior spaces with a topographic survey (TS16 by Leica Geosystems, Rome, Italy), to optimize point cloud alignment results; a GPS survey (GS18 by Leica Geosystems, Rome, Italy), to get absolute coordinates and a UAV photogrammetric survey (Mavic Mini by DJI, Shenzhen, China) for the roofing part. The project of survey was designed to ensure an adequate level of detail for the characteristic elements detected in the previous phase, in terms of resolution, in fact, the integrated survey activities were designed to cover with a $2 \times 2$ cm grid the entire building and get a $1 \times 1$ cm grid for these elements. The third phase, data elaboration, aimed at process and integrate all the raw data coming from the previous phase to get a complete 3D point cloud of the building (Figure 3). This point cloud has been than evaluated to check the global accessibility of all the spaces, the metrical accuracy, the resolution of points, the recognizability of constructive feature of the building, the materials used and the surfaces state of conservation. This complex survey process has been designed beforehand to be applied to the entire Sapienza real estate to guarantee a homogeneous and consistent 3D database able to guide the modelling choices during the next HBIM phase.

### 3.1.3. The "Black Box" Modelling Approach

In this framework, the HBIM of the case study has been approached as a "black box" model. Among the complex system of features that the building carries on, only the tangible aspects, related to the physicality of the object, have been considered. This means getting rid of all those issues connected to the reading of the object through specialistic analysis in terms of geometry, proportion, and interpretation of design intentions.

This choice comes from two main reasons. The first one is connected to a practical aspect, the correspondence between the 3D point cloud and the model is an objective datum that can be metrically evaluated and reported. This can be easily solved in BIM adding a simple and verifiable parameter that becomes representative of a certain correspondence.

The so-called "as-is" BIM, in the AEC industry, is generally interpreted following this approach. The term is sometimes used as synonym of "as built" BIM, it refers to a model which describe the building, from the geometric point of view, in its current state [30]. The second reason is more methodological, it is linked to the interpretation of the model as a digital twin of the object. With this approach, the correspondence between point cloud and 3D model allows specialists such as structural engineers, architectural conservation restorers and energy system designers to carefully plan their interventions on the digital model leveraging the fact that what is represented digitally is fully consistent with the building itself. From this point of view, the entire modelling process looks like the one, consolidated by now, followed for the architectural survey representation process.

In this field, the 3D point cloud is transformed into a set of 2D models–plans, elevations, and vertical sections—derived from an interpretation of the object aimed to recognize the "syntax" of the building [31].

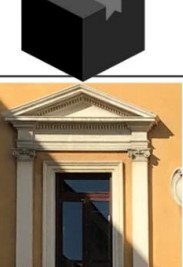

**BUILDING NAME:** Edificio in Piazza Borghese, 9
**BUILDING CODE:** RM050

**BUILDING CATEGORY:** B, stratified buildings of early XX century

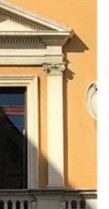

**Description:** window's frame with ionic architectural order and pediment
**ID:** RM050_P03_FIN02
**Historic relevance:** high
**Level of Detail in BIM:** high

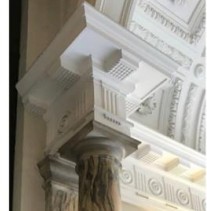

**Description:** entablature
**ID:** RM050_PTE_COR01
**Historic relevance:** high
**Level of Detail in BIM:** medium

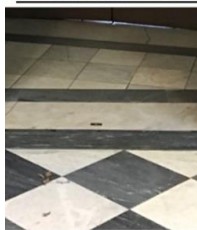

**Description:** marble flooring with white and black squares
**ID:** RM050_PTE_PAV02
**Historic relevance:** medium
**Level of Detail in BIM:** high

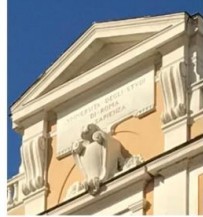

**Description:** plaster pediment
**ID:** RM050_PC_FR01
**Historic relevance:** high
**Level of Detail in BIM:** high

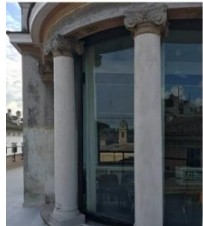

**Description:** ionic architectural order
**ID:** RM050_P03_COL02
**Historic relevance:** high
**Level of Detail in BIM:** medium

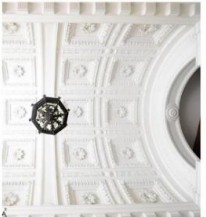

**Description:** coffered barrel vault
**ID:** RM050_PTE_VOL01
**Historic relevance:** high
**Level of Detail in BIM:** medium

**Figure 2.** The building in Piazza Borghese, analysis, and classification of main architectural element.

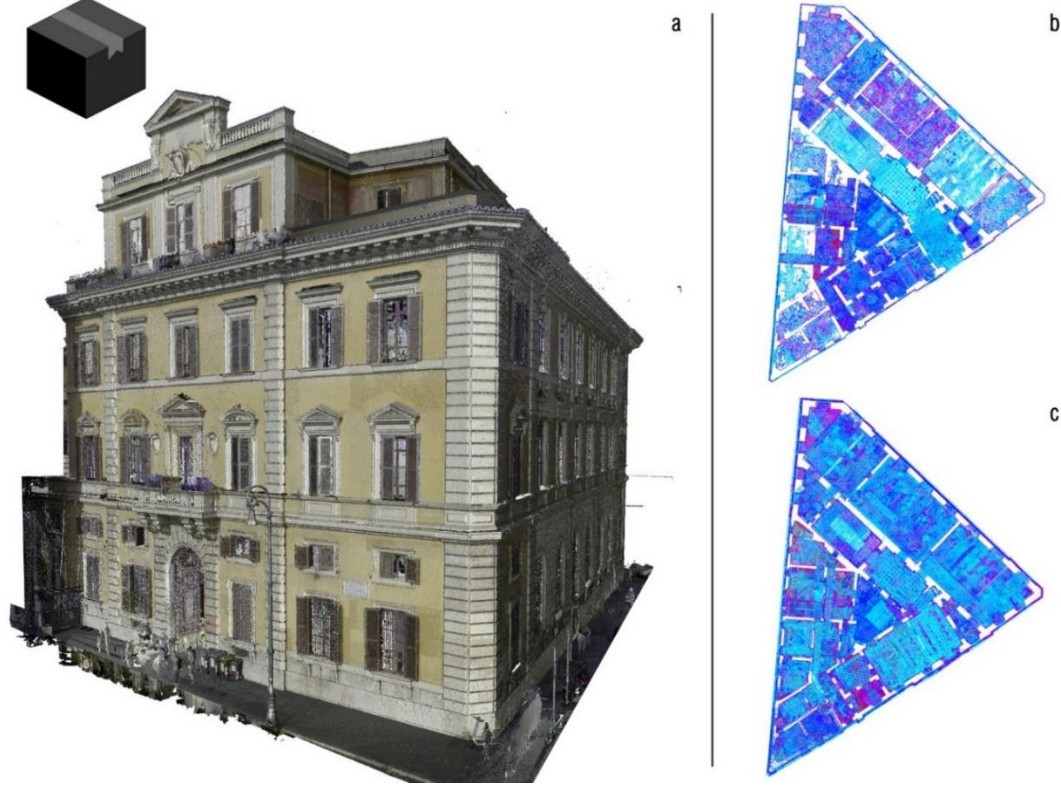

**Figure 3.** (**a**) The building in Piazza Borghese, point cloud. (**b**) Planimetric view of the building ground floor, point cloud. (**c**) Planimetric view of the building first floor, point cloud.

What is than represented through 2D models is the result of this interpretation considering a certain scale of representation with which a certain level of uncertainty is associated. Following this step, specialistic analysis can be performed basing on the 2D models elaborated. In BIM terms, this is translated into a first interpretation phase through which the specialist must detect, once again, the building "syntax" and translate it into the structure of the model. This transformation, as for the 2D models, is guided by a metric reference in terms of adherence between the point cloud and the model. Also in this case, the result of this process is a solid base for the several types of readings connected to the object current state (Figure 4).

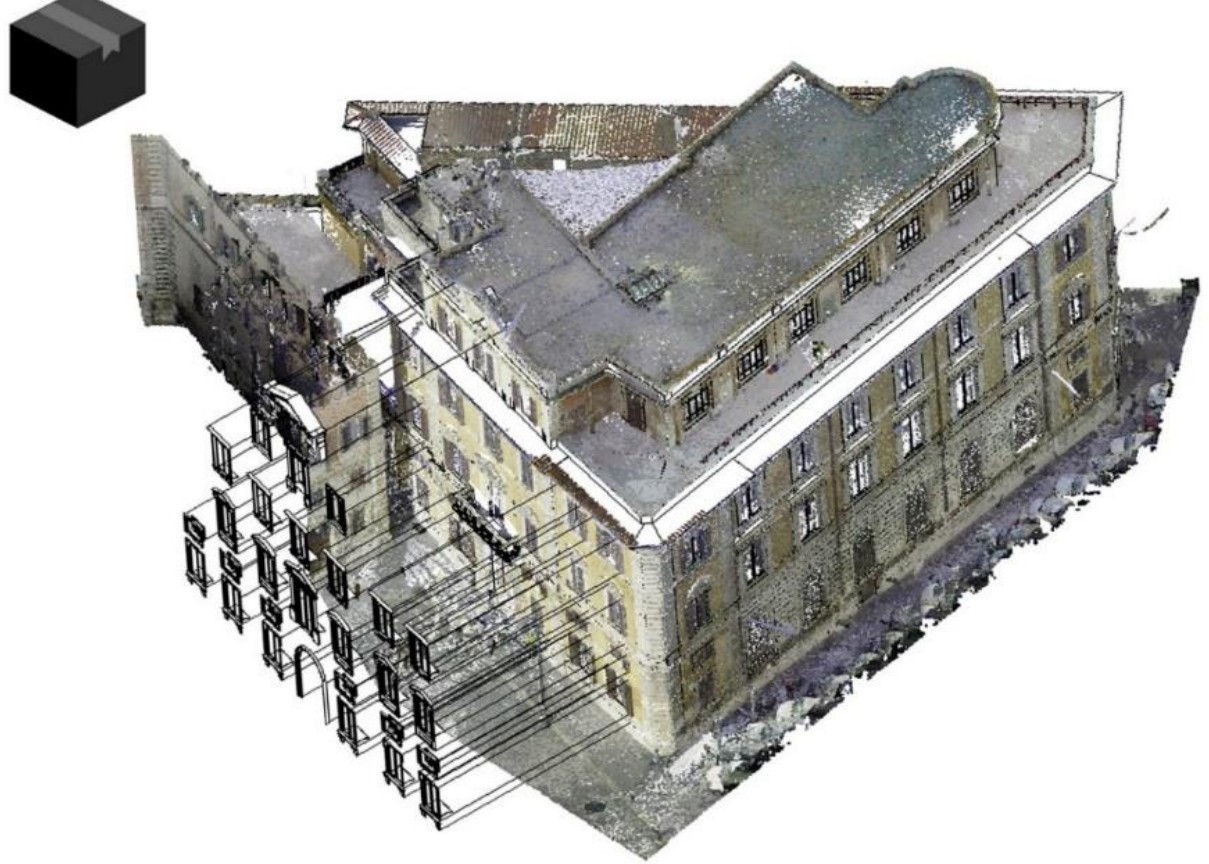

**Figure 4.** The building in Piazza Borghese, "black box" model.

Given this, in practical terms, the model does not follow any predetermined structural grid and, coherently, it does not give for granted any geometric alignment between components. Modelling operations are conditioned only by surveying data, and the only interpretation admitted is the translation of the building element typologies into BIM categories, families, and types. (models discussed in this section have been elaborated using the BIM authoring software "Autodesk Revit, 2022". The terms here used to refer to the model semantic structure are the ones used in Autodesk ecosystem). The Level of Development (LoD) consider the point cloud distinction of the two levels of detail, expressed by the resolution and the metric accuracy (Figure 5). According to this, characteristic elements that were preliminarily detected and captured with a higher resolution, such as the main staircase, the marble flooring of the main rooms, and all the components of the architectural order, have been modelled with a higher geometric accuracy, thanks to the high-resolution point cloud, and a higher level of Information, thanks to the bibliographic research conducted on each of these elements.

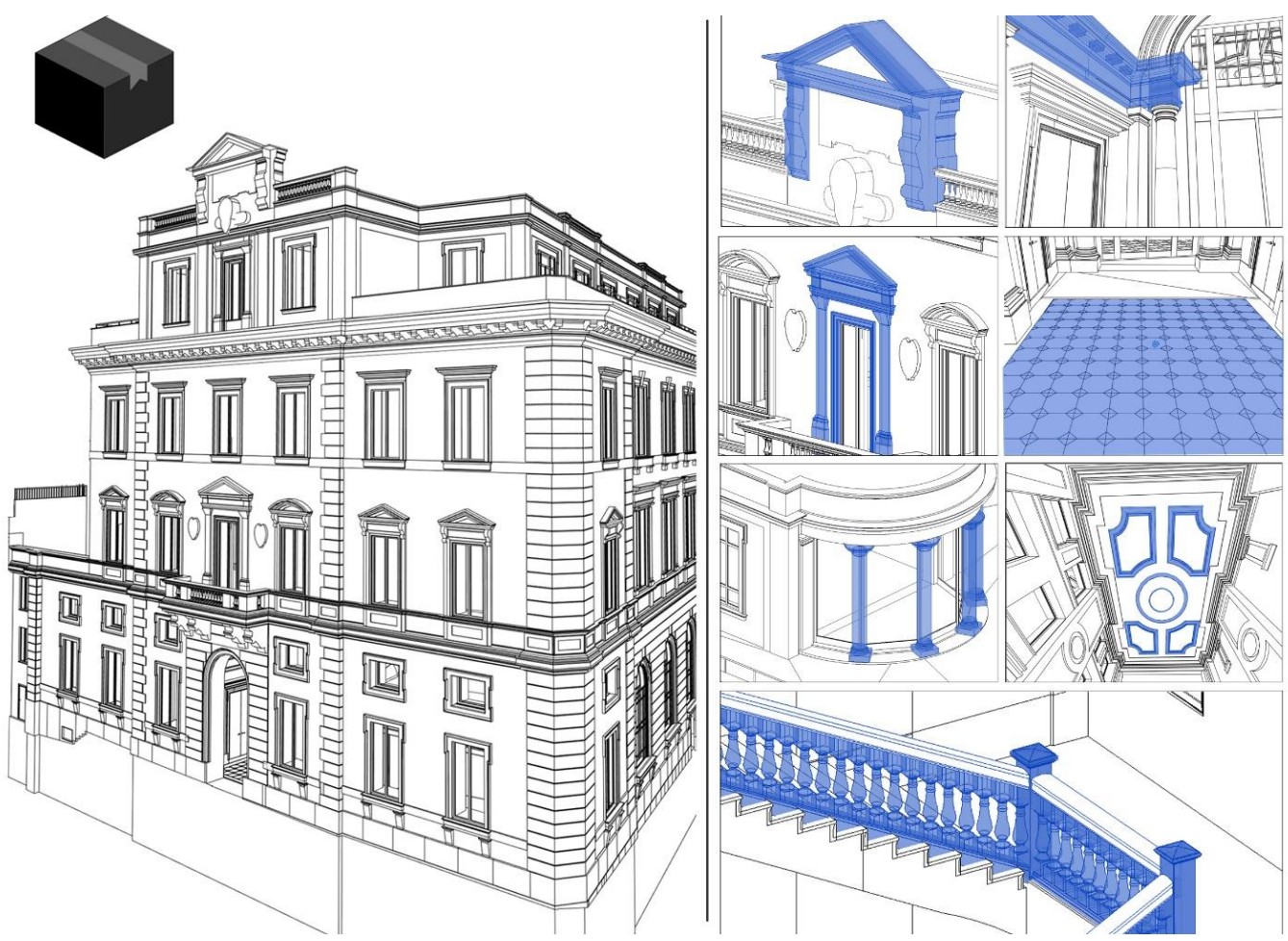

**Figure 5.** The building in Piazza Borghese, the elements of the "black box" model.

### 3.1.4. Fields of Application of HBIM "Black Box" Modelling

The case study of piazza Borghese building shed light on a specific way to approach the HBIM process. Apart from the previously mentioned limitations, the "black box" model is conceived as the digital surrogate of the existing building in its current form. This type of model takes advantage of 3D massive data capture techniques and mainly relies on survey data. Given this, the "black box" model focus on the tangible aspects of the architecture and, for this reason has a specific field of application and way of use. When dealing with HBIM, modelling process is never for its own sake, products are always used with the purpose of optimize the entire building process and, to this scope, further specialistic analysis, in different disciplines are performed on the model itself. For this reason, the modeling approach must be explicit and shared beforehand among all the specialists involved in the process to guarantee a result that can be used in the correct manner. This leaves no space to ambiguities; the specialist must always be aware of what to expect from a specific type of model and what makes sense to ask for. "black box" model can find a good application in terms of building structural analysis, for example, because of the capability to report with accuracy each element localization, this information give space to the study of cinematics occurred on the building or how the constructive anomalies of the elements might affect the global building structural behavior. Similarly, also in the design and architectural conservation field, having a certain level of adherence between the real object and the digital one allows to predict, during the design phase, possible issues related to building irregularities.

### 3.2. "White Box" Modelling of the Botany Institute at Sapienza Campus, Rome

3.2.1. The Building History

The Botany Institute (Figure 6), designed by Gino Capponi and built between 1932 and 1935, represents, together with the School of Math, the most refined and luminous building of all those in the Sapienza campus in Rome. Sapienza University campus is considered one of the most characteristic achievements and significant stages of modern Italian architectural development from both an organizational and technical point of view and from a historical-artistic point of view. The architect Marcello Piacentini chose some young Italian architects to collaborate with him in the design of this large complex buildings. The process started following the desire of the Duce to make the new Sapienza campus the best expression of the Italian architectural and constructive genius of the fascism era. Piacentini organizes a technical office to take care of each project from the general arrangement down to the smallest detail, from the choice of materials, to the choice of the supply specifications, down to the organization of all the technical executive drawings [32].

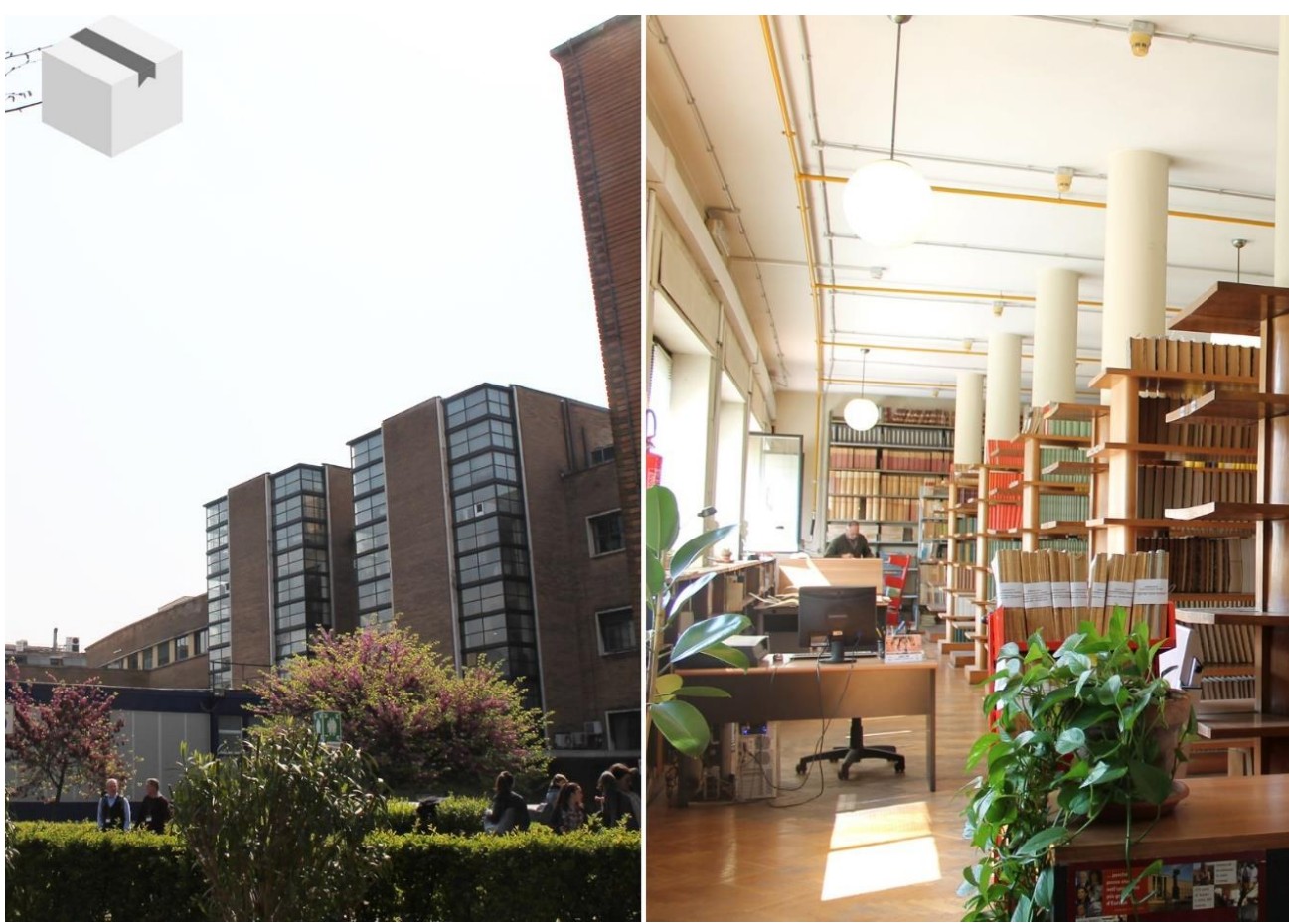

**Figure 6.** The Botany Institute.

The Botany Institute appears as a particular building, characterized by the close dialogue between compact and symmetrical masses and transparent volumes and surfaces. The complex is distinguished from the other buildings of Sapienza campus for the coexistence of horizontal masses and vertical elements, the glass towers. The two glass tower, the deep ribbon windows and the transparent curtain wall of the back façade, demonstrate an architectural sensitivity strongly influenced by European rationalism.

Since 1962, following the delocalization of the Pharmaceutical Chemistry Institute to a new structure built next to the School of Math, the building has undergone some transformations. In the internal distribution some minor changes have not compromised

the formal structure of the original project. The construction of a prefabricated structure, placed in 1969 in the area in front of the entrance stairway, on the other hand, interrupts the symmetry and the elegance of the two-tone marble and travertine of the architectural context to which it belongs [33].

### 3.2.2. Architectural Survey for HBIM Purposes

Since 1962, following the delocalization of the Pharmaceutical Chemistry Institute to a new structure built next to the School of Math, the building has undergone some trans-formations. In the internal distribution some minor changes have not compromised the formal structure of the original project. The construction of a prefabricated structure, placed in 1969 in the area in front of the entrance stairway, on the other hand, interrupts the symmetry and the elegance of the two-tone marble and travertine of the architectural context to which it belongs [33].

The transition between the first aim to the second one shed light to the problems of modelling approach in the complex transition from an uncritical datum, obtained from massive 3D survey to critical elaborations. They require a certain knowledge of the case study to represent its significant elements, in relation to the objectives of the survey. Such an operation shows the importance of knowing the project, its setting, its main characters. The architectural survey, in fact, is "the reconstruction of the monument design" [34]: the scholar reads the work analyzed to understand its static, functional, and formal characteristics, to document them through models. This assumption remains unchanged even in the well-established shift from two-dimensional models to three-dimensional informative models. Given this, an HBIM should also consider this type of specialistic reading.

These assumptions guided the survey of Botany Institute, conducted by integrating 3D capturing methodologies (3D laser scanner C10 and total station TS01 by Leica Geosystems, Rome, Italy, ) (Figure 7). The outer spaces, the facades and the entrance hall were surveyed by a dense network of laser scans, aligned through the topographic survey that allowed both to control metrical uncertainty and to measure characteristic points in the entrance and in some inner spaces. Rooms were then measured by direct survey, due to time constraints and the impossibility of interrupting ongoing activities in classrooms and offices. These data were integrated with those derived from the 3D capturing conducted through UAV systems by the Area Gestione Edilizia office of Sapienza. Numerical models derived from the processing of all available data contributed to set the basis to which information derived from bibliographic research and analysis of archival documents could be integrated (Figure 8). They were essential to have all the information that could not be directly deduced from the survey data or that would require destructive interventions, the stratigraphy of the walls and floors, the detail of the assembly of the technological components, but also to compare the design of the building with its current state.

### 3.2.3. "White Box" Modelling Approach

In this framework, the HBIM of the case study has been approached as a "white box" model (Figure 9). The symmetric configuration of the building and its axiality are well visible even at a first sight, they were analyzed referred to the formal and compositional characters present in Capponi's design.

Reading the design building features is fundamental for a correct preparation of the parametric model following "white box" approach. Right from the project drawings, it is possible to find compositional and aggregation rules analyzing constructive elements of the building volumes. The way these elements are designed and represented through drawings is very close to the one used for contemporary new constructions. This similarity is partly connected to the fact that it is a rationalist building, and it is partly because constructive solutions and materials are pretty similar to the contemporary ones. As with the other buildings on the Sapienza campus, the Institute of Botany project is described through an extensive graphic documentation framing the building from its urban context to the details of design solutions and technological components. The joint collection and analysis of

archival documents and survey data allowed us to define first, the criterion for decomposing the architecture [35], then the approach to modeling. Careful examination of all the available documentation made it possible to extract information on the geometric principles and spatial features of the building, identifying its layouts, the load-bearing elements, the structural axes, the non-structural components, the horizontal and vertical constructive systems, the openings, the vertical connection elements, the exterior finishing materials, interior cladding, and technological systems. These various elements were recognized not only from a geometric and morphological point of view, but in their constitution in terms of dimensions, materials, typology, and use in the more general context of the analyzed project, highlighting, even in the case of existing buildings, the parallelism between BIM processes and construction site practices.

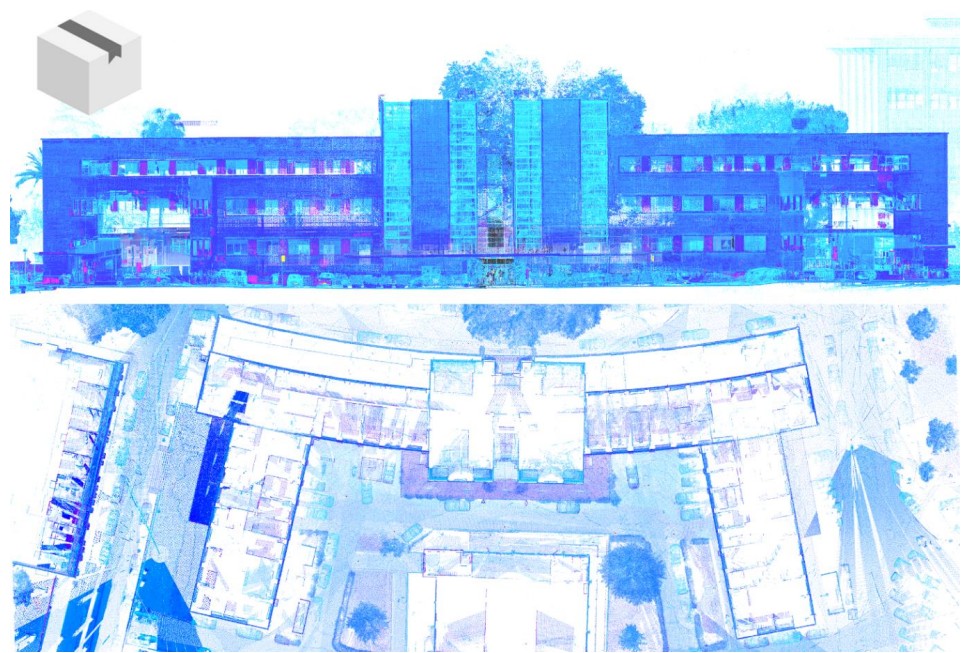

**Figure 7.** The Botany Institute, views from numerical model derived from 3D integrated survey.

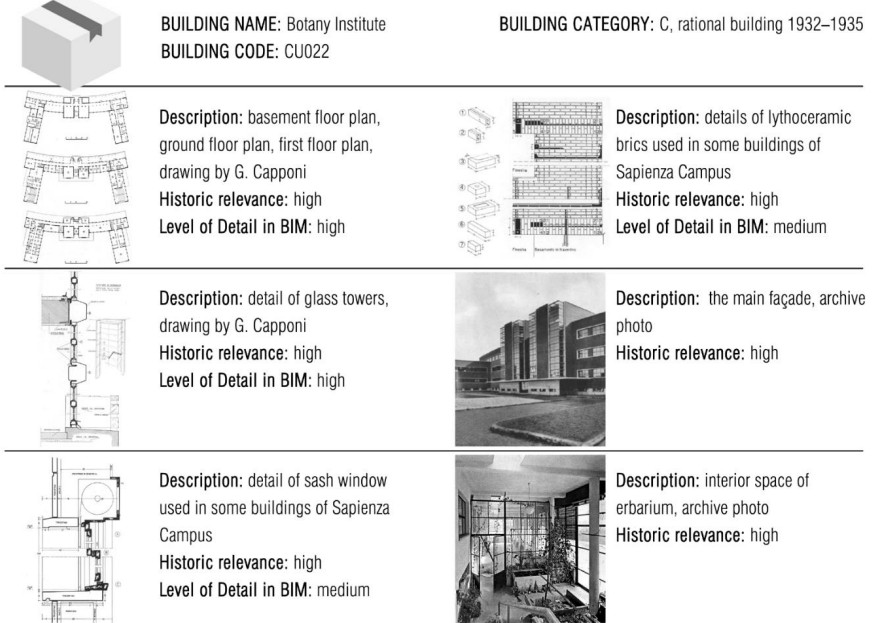

**Figure 8.** The Botany Institute, historical image of the internal space derived from bibliographic research.

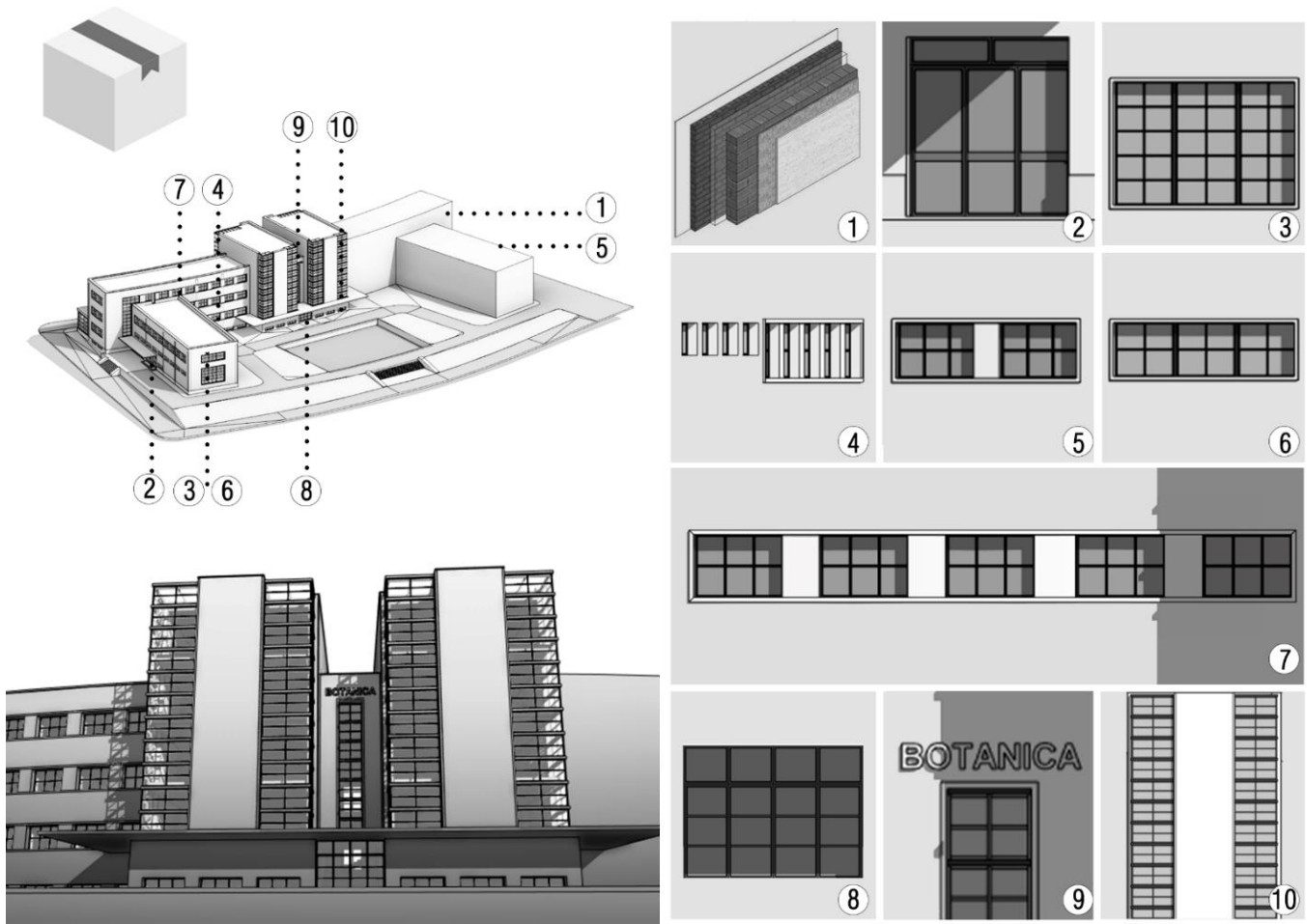

**Figure 9.** The Botany Institute. Detailed view of constructive solutions for external surfaces (**right**) and their localization in the BIM model (**left**).

Textual, graphic, and photographic material, which also details the location of mechanical, electrical and plumbing systems played a key role.

The symmetrical layout of the building, its geometry and repetition of elements were the starting point for a "white box" approach model construction. It followed the same logic as the knowledge operations, starting with the definition of the reference grids to immediately identify the structure of the building, made of concrete pillars, and structural alignments of architectural elements such as the exterior curtain walls, interior partition walls, and openings. Grids, levels and reference planes constitute the geometric rules for the elements' location, which are thus linked by parametric relationships. Therefore, the structure of the model, as here meant, involves a discretization and rectification of the geometries of the elements according to the project.

Pillars, masonry, floors, and stairs are modeled by modifying the construction and dimensional parameters of 3D components already present in the Audesk Revit modeler because they are clearly currently used in the new design. The seriality of the elements, found both on the facades and in the interior solutions, was parametrized within the modeler both for the load-bearing structure, which alternates pillars with different geometries, and for the vertical closures, opaque and transparent, as well as for the bricks and travertine slabs that make up the exterior coating. The available set of information proved to be exhaustive with respect to defining profile geometry, interior stratigraphy, and finishing materials. Component modeling included the customization of digital objects, as they came from a meticulous choice of design solutions appropriately produced for the university campus buildings. The cross-analysis between survey data and archival sources

allowed the definition of some starting parameters and attributes of the digital object, such as the thickness of the walls, the wall structure, the type of exterior coating in travertine romano slabs or litho-ceramic bricks, and the type of the interior plaster finishing. Different modelling solutions were adopted for the exterior coating and transparent components. The former was structured as a modification of the curtain wall family, defined through geometric matrices that regulate the number of elements and the distance between grid axes. Windows of the towers were modeled as a curtain wall, while five types of the windows were constructed through the modification of dimensional parameters of panels and glass. By customizing a single type of curtain wall, modelling process was greatly optimized. The "white box" approach can be here evaluated from different points of view starting from already presented researches [36].

The first concerns the logic that can be used to break into elements the building and choose how to approach the modeling phase. These two phases, both fundamental to understand the building and reconstruct it according to a parametric approach, follow the rational logic of the design of the building. Both the current construction process and the one of the digital model start from the definition of layouts, then the construction of volumes, then the relationship between solids and voids, and coating, considering reference elements from the design stages. The modeling was able to reproduce the project's construction process and the configuration of the elements, which were made according to the principles of serial production, which required their repetition and standardization.

The second consideration relates the actual correspondence between the real object, represented by the numerical model, and the ideal model, consisting of the HBIM, which brings together all the constrains of the original building project (Figure 10). Although the starting point consists of a highly detailed survey, which highlights the features of the building in its general appearance and architectural details, the uniformity and standardization of the project setting prevails over the difference that may be found later in the realization. Specifically, the repetition of architectural components shows variations of few centimeters, so it was possible to make simplifications, considering these variations related to the practical operations of installation, which are never exempt from inaccuracies. This simplification, in fact, corresponds to a metric approximation in the description of shapes and is closely related to the approach followed for this case study. Replicating the "white box" approach to other buildings would probably offer similar results in terms of deviation between the numeric and parametric models. However, currently, unlike in many aspects of the BIM process, there are no fixed parameter to evaluate how acceptable this deviation is, considering that the modelling of existing buildings aims at using the parametric model as an as-built model.

The last point concerns the information needed to build the model. In the case of new construction, it is always possible to know features about the wall stratigraphy and inner composition of architectural elements, because they are decided a priori by the designer. On the contrary, in the case of built heritage, it is not always easy to access such information. This makes the models described in the first case to be homogeneous in the LOD definition; in the second case, on the other hand, it is difficult to establish LOD without having first collected and analyzed the available data and information, making it difficult to know the characteristics that the model will have.

### 3.2.4. Fields of Application of HBIM "White Box" Modelling

The case study of Botany Institute shed light on a specific way to approach the HBIM process. Apart from the previously mentioned limitations, the "white box" model is conceived as the expression of ideal model. This type of model takes advantage of design decisions, from general to detailed scale, which cannot be neglected to achieve the purpose of a deep knowledge. The use of HBIM, in fact, aims at optimizing the management and planning processes of interventions on existing artifacts. In this framework, not only the professional figures for planning works, but also those related to the protection and preservation of the architectural heritage. Specialists such as historians, restorers, and

architects define the operations to be conducted downstream of a deep knowledge of the artifact. Historical research, analysis of archival sources, study of the object from its ideal setting to the transformations it has undergone over time, metrological, proportional and compositional analyses (they are just few kinds of the analyses that can be conducted on an artifact with historical value), are key investigations on which conservation projects rely. Thus, the "white box" model seems to meet these needs, optimizing the concept of HBIM as a database. This approach is particularly coherent for Sapienza campus buildings. By design, all the buildings of the university campus do share the same materials, construction techniques, wall stratigraphy, and technological components. In the idea of tracing the construction process and optimizing the modelling one, these elements could be modeled as general elements, shared, and adapted to the different case studies. This way of managing BIM workflow is particularly promising. In fact, it is possible, in this case to use the same families in the other buildings models in which the same material and construction characteristics or the same elements are recognized, following the homogeneity of the Sapienza campus project. The different models can share the same set of common rules; considering the template on which the model is based not only as a container of rules and tools but, to all intents and purposes, a database to store and manage building construction solutions of the university campus.

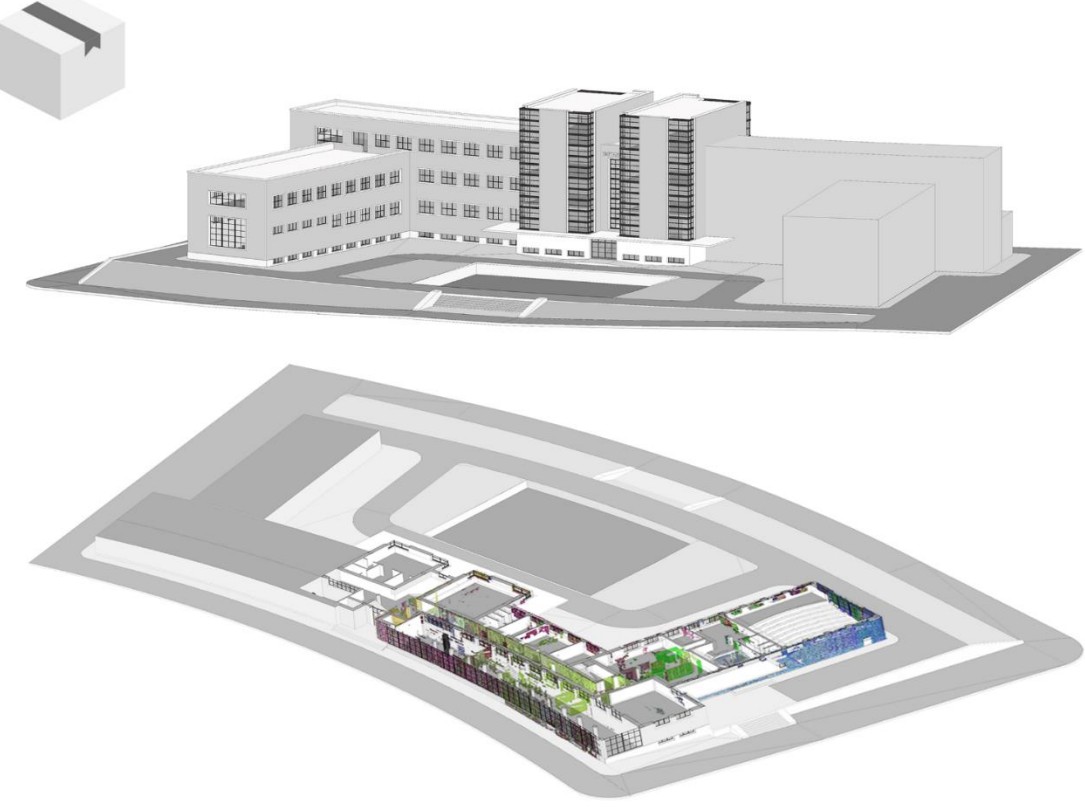

**Figure 10.** The Botany Institute. "White box" model.

## 4. Results

### 4.1. Model Evaluation: Issues and Characteristics of the "Black Box" and "White Box" Approach

The two case studies here presented have highlighted that the final use of the model influences the entire workflow, from the meta-modelling phase to the final product. This assumption implies that, currently, a preliminary evaluation of the use of the model is required to ensure a good adherence between the model real features and the model expectations requested by specialists.

On one hand, the "black box" approach" leads to a good and coherent informative model that can be used as a solid base for all those analyses conducted on the building in

its material consistency; on the other hand, it does not take advantage of parametrization options which are a BIM system peculiarity. In fact, by setting up reference planes, grids and levels to which 3D objects are connected, it is possible to act directly on a series of elements that share the same alignment or that do belong to the same constructive system. For example, having a "black box" approach for modelling a beams system means considering each element individually and disconnected by the others so to be defined by only element-based parameters. This allows to control the interaxle spacing of every single beam reporting every variation from a standard configuration. On the other hand, a "white box" approach would mean parametrize beams interaxle space, beam type properties and beam number for each room. This would mean having the possibility to act directly on the entire system, taking advantage of parameter settings to speed up the modelling phase and ensure a better overall control of the entire model. A similar evaluation might be made on walls alignments using structural grids as a first reference. When dealing with architectural design, the project representation starts with setting up the main axes and proportions of the building, identifying symmetries, geometries, and the reference grid. Given this, when dealing instead with a built architecture, it is natural for the architect's mind to look at the building trying to read its design traces, and detect proportions, symmetries and geometries that guided the project. In the case of the "black box" HBIM, the design intention reading leaves space to what happens just after the designing phase, reporting the irregularities of the building construction and the variations occurred starting since then up to now. This means, once again, getting rid of geometric predetermined references and approach the model documenting the building as it is.

These general principles need then to be tested through selected case study to evaluate limits and effectiveness. On this topic, a first distinction can be done basing on building typology. Considering the Sapienza real estate, we can detect four main categories: stratified buildings, historic buildings conceived and constructed in just one phase, rationalist buildings of the university campus and auxiliary structures built with contemporary technologies and materials. Depending on the building category, the "black box" approach assumes a specific connotation, just to name an example, dealing with a contemporary building, designed, and constructed using prefab components makes it essential the definition of a structural grid before starting the modelling operations and the identification of object types repeated in the structure several times. On the contrary, modelling a highly stratified historic building such as the one in Piazza Borghese, implies in a stronger way the definition of a specific approach to define which path the modelling should follow and witch data it should consider. In this case, the "black box" approach can be fully adopted to extensively describe the real object introducing a limited metric approximation. On this issue, the HBIM "black box" model, even if it is built basing on the point cloud, still reports the same technic constraints connected to the BIM itself, such as the complex management and modelling of complex geometries and the great limitation in the morphology representation when it comes to non-standardized shaped elements. For this reason, the "black box" BIM model is thought to be always connected and integrated with the point cloud so to always keep track of all the geometric and morphologic simplifications occurred during the modelling phase. The "white box" model, instead, can be built even only by referring to the bibliographic references and architect's constructive drawings.

### 4.2. Towards the Digital Twin: The Change of Perspective

In conclusion, the two case studies highlighted two modelling workflows derived from two different meta-modelling approaches. While the "black box" model allows simulations that consider the imperfect processes linked to the object's life cycle, the white model provide an interpretation of the object that make explicit the theoretical aspects underlying the building's design process. In both cases we have a partial model of the building, and, because of this incomplete information, we simply cannot consider them as digital twins.

Given this, it is evident how the construction of the Twin must overcome the limitations inherent in each of the two approaches by fostering inclusive solutions where both modes of

critical re-reading of the data coexist in a single model. A building, in fact, is the expression of a design ideality that assumes, then, a physicality through a process of approximation; both these aspects seem to coexist within the building organism.

The presence of this dichotomy between ideal and real is discordant both from an ontological point of view, due the fact that the real object is just one, and from an applicative point of view, the digital model, the digital twin is one as well. This is the key point still to be clarified.

The following two sections aim to shed light on this issue to provide a possible third way in modelling approach, a "grey box" (Figure 11).

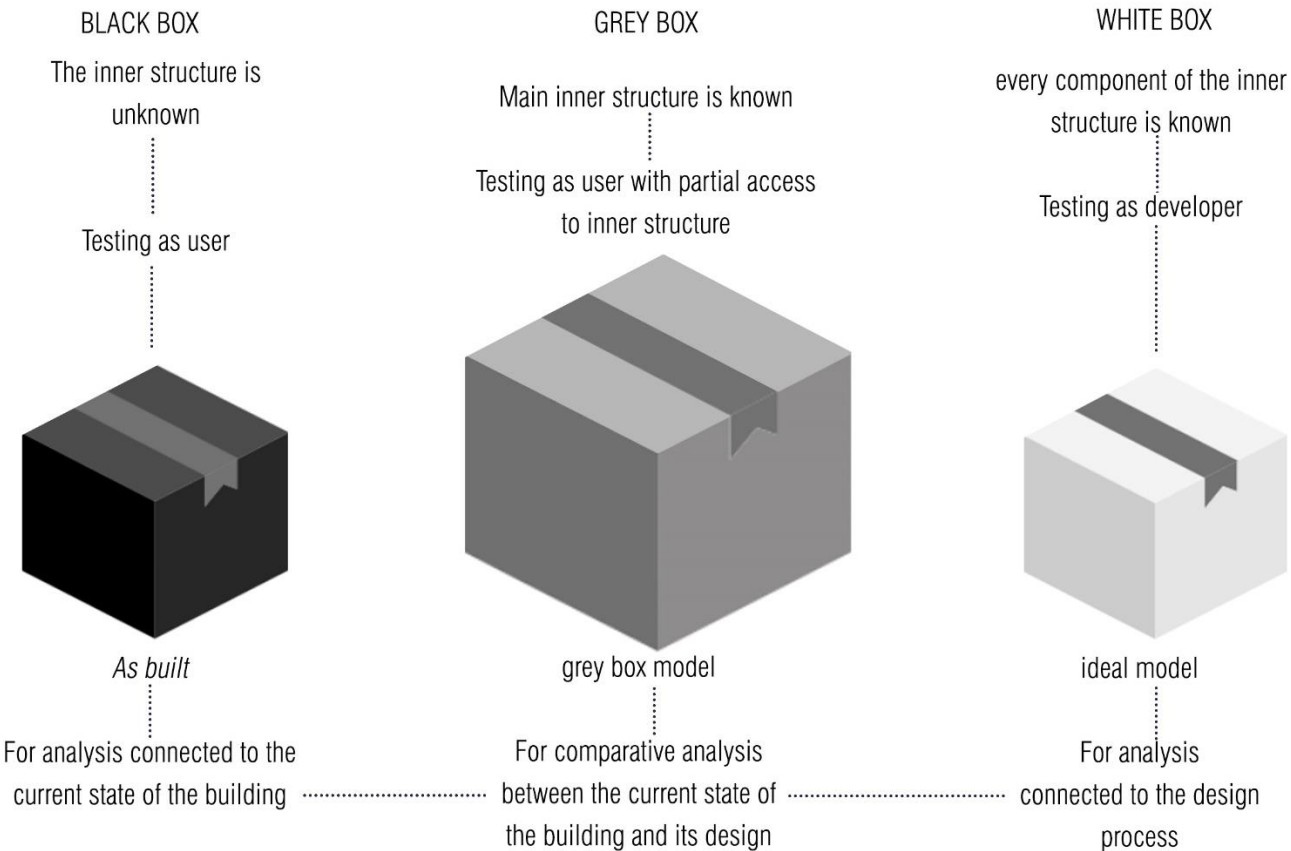

**Figure 11.** "Black box", "white box" and "grey box" approach.

## 5. Discussion

In 1935 Edoardo Persico, one of the greatest theorists of modern architecture, tried to motivate the most profound reasons that move the architectural design process, he concluded a very famous lecture, entitled "Prophecy of Architecture", saying that Architecture is the "Substance of things hoped for" [37]. For years, this sentence fed the Italian architectural debate.

Even if this concept seems to be far from "Information Modeling", the conclusion contains not only the proper sense of Architecture as a complex phenomenon but can represent a clear guideline in all processes aimed at the study and analysis (and therefore also the creation of information models) of any building.

Indeed, the study and analysis of a building cannot be separated from a multi-point of view approach, which often lead to results that are not consistent with each other. On the other hand, this is the only way we have to reach a real knowledge of the building that brings together all the different instances.

This aspect is not exclusively linked to the interpretation of an existing building but is embedded in the entire process from the idea to the construction. Architecture, in fact, stems from an abstract idea and every attempt to approach its materialization can be

considered an autonomous step towards the vain attempt to transform what has been 'hoped for' into 'substance'.

Every step, every transposition into the real world of ideality highlights a loss of information. From the small scale, going down to the large scale, the construction is a continuous attempt to keep the original idea as intact as possible. This is in contrast with the pushes coming from operational aspects, which constantly increase the distance of each result from the initial idea/matrix. When the design takes form in the building process, the construction itself and the limits related to the construction materials, resolve the matrices with an 'imperfect' answer, in a strenuous attempt to cope ideality with the real constructive possibilities.

Starting from this, it is possible to state that each building represents only the best approximation to what was intended, taking into account the fact that the act of materialising the idea inevitably modifies it to meet the complexity of the entire process.

From this point of view, the building represents only the last step in this transformation and, in fact, by modifying our point of view, the same building can give information that pertains not only to its current state but also to its ideal version and to all the phases that have characterized its existence.

The building appears to us as an expression of dualism, between ideal and real, and at the same time in each of the possible intermediate configurations. We face an apparent paradox which recalls another one, used to explain some of the phenomena identified at the microscopic scale by Quantum Mechanics, and which we can describe to understand the potential of this complex and articulated vision of reality.

### 5.1. The "Alive, Dead or Both" Building

In 1935, the famous physicist Erwin Schrödinger created a thought experiment commonly known as 'Schrödinger's Cat Paradox' (https://en.wikipedia.org/wiki/Schr%C3 %B6dinger%27s_cat, accessed on 8 August 2022) to explain the first postulate of Quantum Mechanics, related to the reason why the results of analyses of the same physical phenomenon at the microscopic scale give dichotomous results according to the superposition principle [38]. The experiment proposes to put a cat in a box where a radioactive element, in a defined time, could activate a vial of poison with a 50% probability. From a logical point of view, during the predetermined period, the cat would be in an overlapping state: it could be alive or dead, or rather it could be in both possible states at the same time. The only way to unravel the mystery requires observation by opening the box and only then the cat's state will be revealed. In other words, up to the moment of observation, the cat can be in both states, i.e., all the intermediate linear states, without altering the nature of the system. Although Schrödinger's experiment refers to microscopic systems, from a philosophical point of view it is possible to consider taking its principles for the study of complex macroscopic systems.

Analyzing a building (the system), it appears in the same conditions as the famous cat: until a position is taken or rather an observation point is defined (the box is opened), the building is in all possible configurations. From a certain point of view the building is an expression of its ideal nature, that of the project (it is alive), and from another point of view it is an expression of its own defined and tangible nature (it is dead). Going deeper, and from different points of view we can also say that the building is an expression of its history and evolution (i.e., it is in all the intermediate linear conditions between ideality and reality).

If this hypothesis is valid, it is therefore evident that for the study of a building the key element is not the object itself but rather the point of view from which it is observed. And depending on the point of observation, it is possible to obtain a series of information necessary and sufficient to define, in the form of an informed model, a specific state of it, as in the case of the cat when the box is open.

*5.2. The Digital Time of the Building*

The process from ideation to construction represents a diachronic phenomenon that involves the building structure and makes it evolve in a life cycle defined by clear and defined stages. A great advantage of the construction of computer models is the possibility to consider the temporal aspect, the so-called 4D within the digital processes, i.e., to take into account the temporal evolution of the building by simulating the natural loss of ideality connected to the built environment.

Software that manages these models give the possibility of considering the temporal aspect and therefore considers each individual element in terms of the time of its installation, its transformation, its maintenance and even its death. Analysing a building, we can look at in a precise moment in its 'life cycle' as is the case for a living organism.

This specific parameter has been introduced to consider the transformations that changes the nature of a building artefact over time. Evidently, the first thought goes to the construction phases of a complex buildings, which describe a sequence of operations that condition the final consistency of the building. With the same tool, it is possible to extend this discussion to the entire life cycle of the building, from its construction to its renovation or demolishing.

Moreover, the temporal question becomes a more relevant element when dealing with historical buildings, in which what the building gives us back represents the result of a series of modifications that over the years have seen an idea transform, evolve, decay and renew itself, sometimes without these steps having been planned.

However, if we go into the question of time, it is possible to consider certain key moments that are common to any building, whether new or existing.

If the life cycle of a building is analysed, it is characterised by several phases:

1. Design;
2. Construction;
3. Commissioning;
4. Transformation;
5. Refurbishment or Decommissioning.

Actually, any building we are committed to study can be analyzed in its two specific moments of the life cycle: in its phase 1, the conception phase, i.e., when the building has not yet been constructed, or in a moment between phase 2 construction and phase 5 Renovation/Decommissioning. These two specific moments can be identified by means of "white box" (Ideation) and "black box" (Current State) type models, re-proposing the conceptual problem of what is the grey area where both moments coexist.

*5.3. A Possible Solution to the Twin: The Grey Box*

In this apparently contrasting picture between the representation of the idea and the representation of reality, the need to derive a grey space, the Grey Box, (Table 1) seems to emerge as a novel way to express the complexity, i.e., the dichotomy that is inherent in all building processes. It is possible to imagine, therefore, that the 3D information model of an existing architecture must always carry with it the information linked to the situation, modelled from the survey operations, and that linked to the state of the project, necessary to make explicit the geometric genesis, the compositional logic and the geometric reading of the artefact (Figure 12).

Even if the modelling issues connected to this duality are not new [39,40], this possible solution, which is here depicted only on a meta-modelling level, must find its own application to be further tested and evaluated.

The two models here theorized might coexist in the same three-dimensional space acting on two temporally distinct phases. Their integration would be necessary, at this point, to evaluate the transformations during the construction phase as well as the transformations incidental to the temporal development of the building. This structure constitutes a valid support for the different specialist areas involved in which, from time to time, it will

be possible to decide on which temporal phase of the model to act in a conscious and explicit manner.

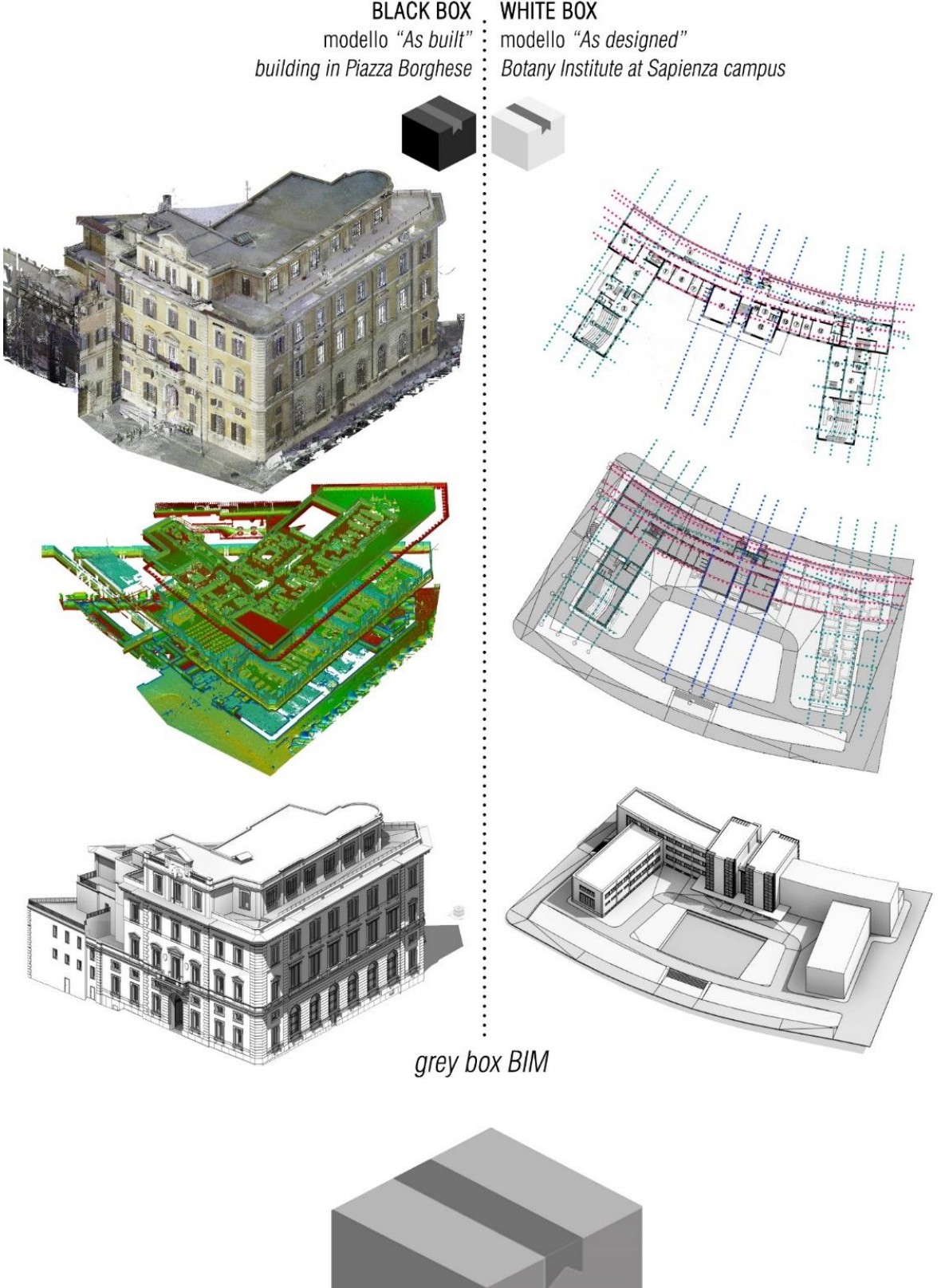

**Figure 12.** Comparison between "black box" and "white box" models.

**Table 1.** Comparison of "black box", "white box" and "grey box" approaches. Checkmarks are located when the modeling approach involves the modeling strategy, X are placed when it doesn't.

| Modeling Approach\Modeling strategies | "Black Box" | "White Box" | Grey Box |
|---|---|---|---|
| Model based on 3D survey | ✓ | X | ✓ |
| Model georeferenced by GPS coordinates | ✓ | X | ✓ |
| Grid and/or reference planes related to 3D survey of the building | ✓ | X | ✓ |
| Grid and/or reference planes related to the original project of the building | X | ✓ | ✓ |
| Structural elements are bound to grids, reference planes and levels | ✓ | ✓ | ✓ |
| Architectural elements are bound to grids, reference planes and levels | ✓ | ✓ | ✓ |
| ARC+STR modeling is referred to central model | ✓ | ✓ | ✓ |
| MEP modeling is referred to central model | ✓ | X | ✓ |
| OUT modeling is referred to central model | ✓ | ✓ | ✓ |
| Different LOD of the elements | ✓ | ✓ | ✓ |
| Knowledge of the constructive solutions adopted | X | ✓ | ✓ |
| Adherence of model to the 3D survey | ✓ | X | ✓ |
| Structural and architectural irregularity | ✓ | X | ✓ |
| Link to archival document | ✓ | ✓ | ✓ |

The Twin must follow the same procedure with respect to both the new and the existing. In this framework, it will consist of a single digital model capable to describe no less than two temporally distinct phases:

1. "white box"—(ideal project)
2. "black box"—(as built).

## 6. Conclusions

As demonstrated, each building might leave the possibility to be reread to find heterogeneous information for specific studies, and consequently, each digital twin should possess so much information that it can be reread from different points of view, different shades of grey.

Hence the hypothesis of solving the complexity of architecture always and only through a dual model, where ideality and reality coexist and where precisely the difference between the two will constitute a first fundamental datum for understanding how time has modified the object of study.

Taking advantage of the possibility of a synchronic construction, of models capable of representing the different phases of the process, from its ideal creation to its imperfect representation, the digital twin will be able to offer the various researchers an adequate amount of information to carry out, each one for what he or she is competent, his or her own deductions or simulations. Operating within the digital world, this approach will provide the basis, in the first instance, for verifying how the object has changed over time and will guarantee a mass of information capable of being exploited, in this case ontologically, by all the specialists in the same way as for the real building.

In this way, and only in this way, it will be possible to overcome the limit given by the attempt to reread the complexity from a single point of view. By operating in this grey area that combines, as happens with reality, what is "substance" and what is "hope", it will be

possible to radically transform the use of these powerful digital tools into a real aid for the study and valorisation, as well as for the management, of the existing Heritage.

Further steps of this research will be focused on applying the meta-modelling principles of the grey box here expressed to a case study. This experiment will highlight applicative strengths and limits of this approach to test the grey box solution on both levels, researches and construction industry ones.

**Author Contributions:** Conceptualization, Martina Attenni, Carlo Bianchini, Marika Griffo, Luca James Senatore; methodology, Martina Attenni, Carlo Bianchini, Marika Griffo, Luca James Senatore; software, Martina Attenni, Marika Griffo; validation, Martina Attenni, Carlo Bianchini, Marika Griffo, Luca James Senatore; writing—original draft preparation, Martina Attenni, Carlo Bianchini, Marika Griffo, Luca James Senatore; writing—review & editing, Martina Attenni, Carlo Bianchini, Marika Griffo, Luca James Senatore. Section 1. Introduction Martina Attenni, Section 2. Background Carlo Bianchini; Section 3.1. "black box" modelling of the building in Piazza Borghese, Rome, Marika Griffo; Section 3.2. "white box" modelling of the Botany Institute at Sapienza campus, Rome, Martina Attenni; Section 4. Results Marika Griffo, Section 5 Discussion, Luca James Senatore, Section 6 Conclusions Luca James Senatore. All authors have read and agreed to the published version of the manuscript.

**Funding:** This research received no external funding.

**Data Availability Statement:** Not applicable.

**Conflicts of Interest:** The authors declare no conflict of interest.

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
