# Peer review of "HBIM Meta-Modelling: 50 (and More) Shades of Grey"

_ijgi, doi:10.3390/ijgi11090468_

Round 1
Reviewer 1 Report
Exploring modeling strategies in the context of HBIM, as well as examining the possibility of connecting in just one digital environment several instances related to the building, are of great importance to practitioners in the field. The authors presented important issues worthy of discussion and comprehensive research examination. However, in order to emphasize the contribution of the article, and for the purpose of understanding its usefulness, authors are advised to refer to the following comments.
1. Introduction
i) After describing the background it is advisable to clearly detail the research questions that the authors are interested in answering, how they intend to do so, and whether they have been given any reference in the literature.
ii) The introductory chapter consists of a number of components that are supposed to be separated. The construction of the introduction as a general chapter, consisting of sub-chapters does not allow for proper follow-up and reference to the questions and the framework proposed by the authors in the study. It is therefore advisable to present an introduction to the research in one chapter and the course of research in another chapter for further understanding.
iii) The chapter describes in detail the construction industry in Italy with reference to Roger's technology adoption curve. If the authors refer in their proposal only to the Italian characterization it should be noted. Beyond that, a bibliographic source should be attached to the specified sociological model.
iv) Lines 254-256. "In other words, if I want to design… , not only I must improve the exchange of information…". Since this is a clarification by the authors it is advisable to phrase it in plural, for further understanding.
v) Since the authors proposed the Blockchain method as protection for the coding of the model, a bibliographic source should be attached and their claim explained.
2. Methodology
i) After the introductory chapter it is not clear how the authors intend to answer the basic questions of the paper and carry out their research. It is advisable to set aside in a separate chapter an accurate description of the methodology and methods chosen. This chapter should include an explanation of why they were chosen.
ii) In the chapter after the introduction there is a mixing of methods and results in a way that does not allow comparison and therefore it is recommended to separate them.
iii) The methodological chapters after the introduction are divided into different approaches, but in the chapter intended for one approach, the second approach was mentioned, in lines 410-411, even before it was presented. It is advisable not to compare approaches before they have been fully presented. Comparing approaches should be done after understanding their characteristics.
3. Results
i) It is recommended to change the chapter subtitle to avoid grammatical errors and not to confuse the presentation of the results with a discussion of them and a recommendation for a third approach.
ii) It is recommended not to present a background in the results chapter but to refer to the various test cases presented.
4. Discussion
i) The authors presented a thought experiment relating to quantum mechanics. It is advisable to present the thought experiment as a kind of image with a bibliographic source citation. The use of this example may have philosophical or physical meanings, so it is important to elaborate.
ii) The authors presented a table comparing the different approaches according to their proposal in relation to a third approach. It is important to point out the practical implications of this choice, as part of the discussion of the comparison results. It is advisable to present a literary source that supports this approach.
iii) The paper lacks a concluding chapter, to emphasize its aims. Authors are advised to present the added value of the third approach alongside its shortcomings. Given the complexity of the approach they have proposed, authors should indicate whether they see any limitations to the study.
Author Response
Thank you for your comments and suggestions. You find here attached a point by point explanation of all the editings we made on the manuscript.
Best regards

Reviewer 2 Report
This paper describes modeling strategies for HBIM, which is a topic of great interest. The methodologies are compared with the black box testing and white box testing from computer science. As presented by the authors, the black box concludes to an “as build” model and on the other hand the white cox concludes to a “as designed” model (Figure 12).
The differentiation from the black to the grey and the white box, as presented in the paper, is related to level of knowledge of the structure of the building (Figure 11). This aspect should be intergraded in the title, since the current one could be misleading. The 50+ shades of grey boxes are not justified in the text. There are 3 main modeling approaches presented (black, grey and white). Furthermore the HBIM meta-modeling should be change to HBIM-Modelling.
The authors in the abstract state that they aim to “identify at what extent the final use of the model might affects, or should affect, the modelling approach itself”. But this information is missing from text. A table showing the link of different final uses of HBIM models with the modeling approaches should be included.
Moreover the evaluation of the modeling approaches as described in Table 1 is not adequate. Table 1 as it is, is more a comparison table than an evaluation table. A set of evaluation criteria should be defined and applied to the 3 modeling approaches.
The novelity and contribution to original research should be described clearly in the text.
The introduction section is very interesting to read. Further and more recent state-of the art references on HBIM modeling are required.
Some more specific remarks:
Line 126: Do you mean material instead of fabric?
First paragraph of 1.1: Define some software for scan to BIM processing.
Second paragraph of 1.1: Quality and evaluation of point clouds are vastly presented in the literature.
Line 226: Add reference for Rogers curve
Lines 330-340: Add some specs for the laser scan
Line 374: Please explain how plans, elevations, and vertical sections were derived from an interpretation of the object aimed to recognize the “syntax” of the building. Was is a manual automatic or semi-automatic procedure?
Please check captures and their references in the text for figures 7 and 8.
Figure 7 is a numerical model or a point cloud?
Author Response

(The authors gave the same response as above.)

Reviewer 3 Report
The manuscript presents 3 different HBIM modeling approaches that the authors name as black box, white box and gray box (a combination of the above). All methods are presented in detail in terms of modeling approach , from reading the building materiality (geometry and morphology), survey of its current condition, definition of component detailing levels.
They claim that the black box method, has to be always connected to the point cloud, due to the model having the limitation in representing the complexity of the represented building.
The second method, white box, was applied to a more recent rationalist building that had original documentation close to the current structure. The geometry and repetition of elements allowed the white box approach to be applied. The authors argue that the white box approach is suitable for buildings whose decomposition and reconstruction process leads to an ideal model, whose components can even be used in other similar buildings
The authors call the gray box the Digital Twin.
The text is well written with few grammatical errors.
The discussion gives an interesting overview of the path towards the Digital Twins for heritage buildings.
Figure 12 is not clear enough, to explain the gray box. Missing an explanation of the proposed Gray box approach
The Botany Institute already had a similar analysis in two other papers, ¨Regenerative Design Tools for the Existing City: HBIM Potentials¨ (14) and ¨Informative Models for Architectural Heritage¨(NOT IN THE REFERENCES, ONLY A OTHER VERSION -22).
The authors should highlight the contribution of these works to the manuscript in question.
Author Response

(The authors gave the same response as above.)

Reviewer 4 Report
The paper discusses models although it seems only representational models are considered (Eppler and Burkhart). BIM data can be visualized in many forms. The nature of the codes used to create building could have been further discussed. Concepts of Pattern Language Christopher Alexander could have been mentioned. The term digital twin and twin need further clarification. Digital Twin can refer to real time monitoring models. A framework for linked data based heritage BIM Phd by Mathias Bonduel covers this. There is a taxonomy of digital twins. Concepts of level of certainty of BIM objects could have been elaborated on. Several areas references were needed such as Rogers curve or Blockchain. Several spelling mistakes buildign and Gray. Future research in this area could be suggested. The research methodology on which the paper is based is not defined. According to Numerical Modeling of Masonry and Historical Structures there can be Block based models, continuum models, macro element models and geometry models these concepts could also be mentioned. You could consider granularity in more detail as covered in Generative HBIM modelling to embody complexity (LOD, LOG, LOA, LOI): surveying, preservation, site intervention—the Basilica di Collemaggio (L’Aquila). These are just suggestions which is why I have suggested a major rewrite although the premise of the paper is sound and the examples given sufficient to demonstrate the concept.
Author Response

(The authors gave the same response as above.)

Round 2
Reviewer 1 Report
The authors made many changes from the original version, which contributed to better understanding. However, a certain revision is required in the discussion chapter and the conclusion chapter, in terms of references and content.
The discussion chapter included a reference to the quantum mechanism, but again did not present a clear explanation regarding the meaning, whether it is philosophical or physical, nor a bibliographic source.
The title of the conclusions chapter should be in accordance with the other main titles, without interpretation. This chapter should not include further discussion, comparison table and accompanying interpretations. It is advisable not to present a second discussion in this chapter, and instead emphasize the limitations of the study and its advantages
Author Response
Dear reviewer thanks for your considerations.
The discussion chapter included a reference to the quantum mechanism, but again did not present a clear explanation regarding the meaning, whether it is philosophical or physical, nor a bibliographic source.
We added few explanation lines (798-800 line). We added also a bibliographic reference [38] and a link to an explicative image of the experiment
The title of the conclusions chapter should be in accordance with the other main titles, without interpretation. This chapter should not include further discussion, comparison table and accompanying interpretations. It is advisable not to present a second discussion in this chapter, and instead emphasize the limitations of the study and its advantages
We changed the title of conclusion chapter and moved the further discussions and comparison tables here previously included to the discussion chapter as an additional paragraph (5.3 A possible solution to the twin: the grey box)
Reviewer 2 Report
The paper has been significantly improved with the authors new additions and corrections.
2-3 lines should be included in the introduction, describing clearly the novelity and contribution of the paper to original research.
Author Response
Thank you for your further comments!
We have added few lines about research novelty from 44-46 lines
Reviewer 4 Report
Although I have stated minor corrections I am happy with the paper in its current form. It provides a well explained way of thinking which is a useful way of framing the subject area. There are those who would consider the concept of Ditigal Twins morphed from the concept of Mirror worlds developed by David Gelernter. Just focusing on graphical models is an acceptable stance. But if this is the stance you are taking it would be good to clarify that at the outset. But I would suggest the concept of black box and white box and combine models relates equally to non graphical models and representation which are an equally important part of the BIM domain. The term BIM relates to models, models are creations that allow insight and understanding. The question is what form of representation (graphical or non graphical) gives the greatest insight or clarity in support of the decisions that need to be made or the results chains that need to be instigated. There are papers discussing level of certainty which are coming at the problem from a different perspective than the black box white box approach. If we have a wall a roof etc in a historic building how do we know what it is. You have used the black box white box approach to get a better understanding. Its a bit like the concept of triangulation or the use of multi method research as a better way of validating knowledge. But we can also numerically weight our objects with an indication of how certain we are about the correctness of the data attached. In the same way to can attach risk and if we are say BIM is IFC's and of the IFC field types to our object. Of course we can extend the data sets if we use produces like "co builder" or "go bim". Actually knowing the inner materials and construction of historic building is often difficult without intrusive investigate. Often we use comparitive buildings but sometimes builder took shortcuts using the wrong materials and constructing their builder for outward appearance but not necessary constructing them in terms of the best building practice at the time. Maybe adding the concept of level of certainty to this work or could be left out although I do thing the integration of the concept could form the basis of another important paper in this field. Another paper to perhaps consider is the integration of semantic mapping into the concepts you have already developed. Hope that helps.
Author Response
Thanks for the interesting observations and considerations! We will definitively take your suggestions into account for further applications of our study.
We added for now just few lines about the issue of the level of certainty (line 328-329), also inserting a bibliographic reference. The topic of the level of centanty/accuracy/reliability is very interesting for us and we, as a research group, are currently conducting researches on it.
Thanks again